# Does Nepal Have the Agriculture to Feed Its Population with a Sustainable Diet? Evidence from the Perspective of Human–Land Relationship

**DOI:** 10.3390/foods12051076

**Published:** 2023-03-02

**Authors:** Ying Liu, Yanzhao Yang, Chao Zhang, Chiwei Xiao, Xinzhe Song

**Affiliations:** 1Institute of Geographic Sciences and Natural Resources Research, Chinese Academy of Sciences, Beijing 100101, China; 2College of Resources and Environment, University of Chinese Academy of Sciences, Beijing 100049, China; 3Key Laboratory of Carrying Capacity Assessment for Resource and Environment, Ministry of Natural Resources, Beijing 101149, China; 4Faculty of Geography, Yunnan Normal University, Kunming 650500, China

**Keywords:** supply–demand balance, land carrying capacity, household questionnaires, sustainable diet, Nepal

## Abstract

Nepal is one of the least developed countries in the world, with more than 80% of the population engaged in agricultural production and more than two-fifths of the population still living below the poverty line. Ensuring food security has always been a key national policy in Nepal. Using a nutrient conversion model and an improved resource carrying capacity model as well as statistical data and household questionnaires, an analysis framework for food supply balance is developed in this study, which quantitatively analyzes the balance of food supply and demand in Nepal from the perspectives of food and calories during the period 2000–2020. Nepal’s agricultural production and consumption have increased significantly, and the diet has been relatively stable over the past two decades. The diet structure is stable and homogeneous, with plant products occupying the absolute position in overall dietary consumption. The supply of food and calories varies widely from region to region. Although the increasing supply level at the national scale can meet the needs of the current population, the food self-sufficiency level cannot meet the needs of the local population development at the county level due to the influence of population, geographical location, and land resources. We found that the agricultural environment in Nepal is fragile. The government can improve agricultural production capacity by adjusting the agricultural structure, improving the efficiency of agricultural resources, improving the cross-regional flow of agricultural products, and improving international food trade channels. The food supply and demand balance framework provided a reference for achieving balance between the supply and demand of food and calories in a resource-carrying land and provides a scientific basis for Nepal to achieve zero hunger under the framework of the Sustainable Development Goals. Furthermore, development of policies in order to increase agricultural productivity will be critical for improving food security in agricultural countries such as Nepal.

## 1. Introduction

Almost one-fifth of the global population lives in South Asia, where most countries have high population density and relatively low rates of economic growth [1]. Agriculture has always had an important economic position in Nepal. In order to feed their rapidly growing population, agricultural resources are under enormous pressure in the region, and the food supply cannot meet the needs of population development [2]. As a result, food insecurity, linked to poverty, persists [3]. A large number of studies have confirmed the importance of food insecurity in South Asia [4,5,6,7]. Poverty is considered to be an important contributor to, and a key consequence of, hunger and food insecurity. In particular, the number of poor people can be used to measure the state of food security in a country [8]. Food security has long been defined as the supply of food that meets the daily needs of all people, and the most important element is the supply and access to food [9]. By the 1996 World Food Summit, the definition had expanded to “people having physical and economic access to safe and nutritious food to meet their dietary needs for an active and healthy life” (2020). As one of the key factors in promoting economic development, food security plays an integral role in maintaining social stability and achieving national autonomy, especially in developing countries [10]. Indeed, agricultural environments in most South Asian countries are highly vulnerable to human or natural factors such as extreme weather events, natural disasters, resource constraints, labor shortages, and armed conflicts [11,12,13,14]. Since the soaring food prices in 2008, how to provide food to the world in 2050 also has been a concern of scholars (FAO) [15]. What is more, since 2019, the world has been affected by the COVID-19 pandemic, food supply disruptions, and lack of income; the number of hungry people has also increased; and food insecurity in developing countries has continued to increase, thus exposing the vulnerability of the world’s external food supply [16,17]. Ensuring food security and maintaining sustainable agricultural development are increasingly focus of scholars [18,19].

In general, the factors affecting food security can be measured from two different aspects, one is the analysis of food availability, access, and use from the individual and/or micro level and the other is the analysis of food supply and demand at the national and/or regional scale [20]. Rapid population growth has led to a steady increase in the demand for food, which is even more pronounced in developing countries where food production may not be able to meet the demand for food based on existing arable land, due to environmental problems, urbanization, and increasing water scarcity [21]. Therefore, many scholars have conducted studies on food security from the perspective of food supply and demand [22,23,24]. Food supply refers to the quantity, variety, and quality of food that a unit can consume at will [25]. Among them, the balance between food supply and demand means that the amount of food supply can meet the basic needs of regional population development [26]. Food supply and demand balance research focuses on trend and forecast analysis. Some studies analyze the relationship between regional food production and food demand and find that the lag between them leads to an imbalance between food supply and demand [27]. There are also studies about the relationship between food supply and demand and food price fluctuations, analyzing the trend of food price changes and making suggestions for ensuring food security [4,21]. Researchers analyze the current situation and future trend of food supply and demand balance through different perspectives, such as food availability, diet structure, food circulation and trade, arable land potential, and farmers’ behavior [28,29,30].

To date, research methods related to the balance between food supply and demand are mainly qualitative or quantitative analysis from the aspect of variety, total quantity, or region. The main methods used include: (1) using the analytical hierarchy process, comprehensive index system, entropy method, and other relative numerical index system evaluation methods for agricultural resource analysis [29,31,32,33]; (2) using the absolute numerical index methods such as the water pollutant concentration, the original ecological food carrying capacity, the ecological footprint, the minimum arable land area per capita, and the population carrying capacity to analyze and evaluate the supply level and the spatio-temporal characteristics of agricultural products [34,35,36]; and (3) based on the comprehensive index system methods such as dietary nutrition supply standards and food demand forecasting models, the regional balance of food supply and demand, the nutritional structure of the residents’ diets, and the resource and environmental effects of food consumption were analyzed [21,37,38].

Nepal is a landlocked developing country in South Asia dominated by an agricultural economy, which generates about one-third of its gross domestic product (GDP). As far as the authors know, the relatively stagnant performance of Nepal’s agricultural sector is mainly due to the country’s vulnerability to man-made and natural disasters, severe climate change, and limited land/production resources resulting in low crop yields and post-harvest losses [3,4,39]. The 2011 Nepal Demographic and Health Survey by the Population Division Ministry of Health and Population, Government of Nepal, found that despite high rates of poverty reduction due to factors such as urbanization and rising income from migrant exodus, Nepal continues to suffer from severe malnutrition, food insecurity, and poverty [40]. In particular, the Government of Nepal has prioritized nutritious and sufficient food throughout the year as one of the fundamental rights of its citizens in the constitution and through the Right to Food and Food Sovereignty Act [41]. More importantly, in line with its commitment to the Global Sustainable Development Goals (SDGs), Nepal has also prioritized food and nutrition security as a key sector dedicated to eradicating hunger and malnutrition, increasing agricultural productivity, and ensuring sustainable food production systems.

Land is the basis of food production. With the growing contradictions between land, food, and population, the question of whether land production capacity and food demand can be balanced has increasingly become a focus of research [42,43]. Among them, obtaining the number of people that existing land resources can support has become a key element in analyzing the balance between food supply and demand. The research mainly focuses on the issues of internal food supply in Nepal and analyzes the changing characteristics of food supply and demand balance in Nepal from 2000 to 2020. Combining with the Food Balance Sheet of FAO, using the nutrient conversion model and improved resource carrying capacity model, the spatial and temporal variation characteristics of food supply and demand levels in Nepal are analyzed from the national, regional, and county scales. The study has two objectives: (1) to establish a framework of food supply and demand balance, examining the food supply and demand situation in Nepal from both the basic human needs and comprehensive demand aspects; (2) to assist the Nepalese government in identifying regional agricultural development differences and formulating more appropriate regional agricultural development policies according to local conditions.

## 2. Materials and Methods

This study is based on data on the production and consumption of agricultural products in Nepal. According to the two standards of the basic cereal supply level (basic demand) and the calorie consumption level (comprehensive demand) and combined with the concept of carrying capacity of land resources, a model of land carrying capacity (LCC) was constructed to measure the level of food supply and demand in Nepal (Figure 1). Among them, the actual calorie supply is calculated taking into account different types of food use, waste, and edible proportions. The analysis is carried out according to different research scales, i.e., from country and geographical region to county.

### 2.1. Data Sources

The structure of agricultural production in Nepal is relatively simple, and thus agricultural products are mainly divided into plant and animal products in this study. Vegetal products are divided into eight categories, including grains, roots, sugar crops, oil crops, pulses, spices, vegetables, and fruits. Animal products are divided into three categories, namely meat, eggs, and milk. Agricultural production data are mainly from the Food and Agriculture Organization of the United Nations (FAO) website (FAO; https://www.fao.org/home/en, accessed on 10 September 2022), the Food and Agriculture section in the Statistical Yearbook Nepal (2013–2019), and the Statistical Information on Nepal Agriculture (2019/20) [44,45]. Specific sources of data on agricultural products are shown in Appendix A. It should be noted that this study used the same statistical range as FAO.

Data on food consumption were obtained from “Food supply (kcal/capita/day)” section of the FAO food balance sheet, and the questionnaire on food consumption of residents from Nepal. Based on the understanding of the residents’ food intake, the Nepalese Residents’ Food Consumption Questionnaire conducts preliminary research of the contents of the questionnaire by sorting the relevant literature and summarizing the previous research experiences and following the principles of scientificity, systematicity, independence, and comparability. After the design, experts from Nepal were invited to discuss and form the final survey questionnaire. The questionnaire mainly includes the following basic contents, i.e., family residence, number of family members and age structure, and main food types, as well as the type and quantity of food intake for three meals a day. A questionnaire survey was conducted in Nepal in December 2021. A total of 200 households’ food consumption was counted, and the validity of the questionnaire reached 100%. Data obtained from the questionnaire were analyzed using Reliability analysis and Factor analysis in IBM SPSS software, and the number of family members, caloric intake of rice, vegetables, meat, eggs, fish, and milk were selected for further analysis. The results showed that the reliability (0.769) and validity (0.779) passed the test, and the quality of the questionnaire results was high, which proved that the results were usable.

National population data are from Area and Population section of the Nepal Statistical Yearbook and FAO. However, the Central Bureau of Statistics of Nepal conducts a national census every ten years, and the results of the 2020 census have not been released. Therefore, county-level population data in 2020 comes from WorldPop (https://www.worldpop.org/geodata/summary?id=49839, accessed on 15 September 2022). The grid data were divided into counties via the county vector boundary using the ArcGIS (10.x) software. According to the total population of the country in 2020, the county-by-county conversion is conducted in equal proportions. This ensures that the sum of the population of the counties is equal to the total population of the country.

The three major ecological divisions of Nepal are mapped with reference to *Nepal: A Country Study* [46] and are combined with district administrative divisions. 2020 land use data from GlobeLand30 dataset (http://www.globallandcover.com/home_en.html, accessed on 30 September 2022). Land area data are from the Statistical Yearbook Nepal 2019 [44].

### 2.2. Food–Calorie Conversion Model

Foods varies by calorie content, and a consistent measurement of food supply and demand levels is realized using the food–calorie conversion model:$$Energy=\sum E_{i}\times Calorie_{i}$$
where *Energy* is the calorie supply level; $E_{i}$ is the *i*-th category of food (Table 1); and $Calorie_{i}$ is the calorie contained in the *i*-th category of food. To estimate calorie intake on the consumption side, based on food production as a data source, a calorie supply model was constructed by comprehensively considering food use (conversion factor), food waste (waste coefficient), and food edibility (intake part coefficient). The model has matching parameters for each food (Table 1). Table 1 provides food–calorie conversion parameters for major categories of food. Food calorie data from FAO Statistics (http://www.fao.org/infoods/infoods/tables-and-databases/asia/en/, accessed on 6 September 2022) matched 180 identified food items. The food conversion factor, intake part coefficient, and waste coefficient in distribution and consumption are mainly from *Global Food Losses and Food Waste* [47]. Referring to the World Bank’s classification of national income groups, the corresponding parameters are selected according to the level of economic development and the geographical region where Nepal is located. For detailed parameters of other foods, please see the Appendix A and Appendix B of the paper.

### 2.3. Definitions of Food and Calorie Demand Levels in Nepal

There are differences in recommended nutrient intake standards for different populations and different living standards. According to the human caloric requirements in the relevant FAO and WHO reports, the caloric requirements of an adult individual weighing 70 kg are 2450–2730 kcal/day [48]. The Lancet Health Commission recommends a caloric intake of 2500 kcal/day. Ceren and his colleagues believe that global calorie demand has been increasing over the past 50 years, reaching 2370 kcal/person/day [49].

This study determined food and calorie requirements for Nepal based on food supply data from the FAO Food Balance Sheet. In particular, only primary agricultural products are considered in terms of food supply, while aquatic products are not included. It has been calculated that per capita food consumption in Nepal in the past five years is 403 kg, and calorie intake per capita is 2425 kcal. Therefore, the cereal per capita and calorie standards were 400 kg/year and 2400 kcal/day, respectively.

### 2.4. Food Demand–Supply Balance Model: Land Carrying Capacity Model

Combined with the concept of carrying capacity of land resources, the LCC model was constructed and improved. It can measure the number of people in Nepal that can be sustainably supported by regional arable land resources under a given level of food consumption. The model is calculated according to two aspects of meeting the basic and comprehensive demands of human survival. Model results are used to measure the state of regional food supply and demand. The LCC and LCCI was analyzed in relation to cereal demand and calorie requirement based on the characteristics of food production in Nepal using the following model:$$LCC=\left\{\begin{array}{c} LCC_{c}=G/400\\ LCC_{E}=E/2400 \end{array}\right.$$
$$LCCI=\left\{\begin{array}{c} LCC_{c}I=P/LCC_{c}\\ LCC_{E}I=P/LCC_{E} \end{array}\right.$$
where $LCC_{c}$ is LCC estimated against cereal demand; $LCC_{E}$ is LCC estimated against calorie requirement; G is cereal production; E is calorie supply; P is current population size; LCCI is LCC index; $LCC_{c}I$ is estimated relative to cereal demand and measures the degree of cereal supply–demand balance; and $LCC_{E}I$ is estimated relative to calorie requirement and measures the degree of calorie supply–demand balance. LCCI values are classified into three levels and six ratings (Table 2) to describe the food demand–supply balance level.

## 3. Study Area

### 3.1. Geographical Environment in Nepal

Nepal is a landlocked country and borders China to the north, and India to the south, east and west. It is commonly divided into three ecological divisions, namely the Mountain belt, the Hill belt, and the Terai belt, according to decreasing height above sea level (asl) (Figure 2). The Mountain region is mainly north of the northern mountain region above 4000 m asl and represents the central part of the Himalayas. The Hill region is located in the southern part of the mountains, mostly between 1000 and 4000 m asl. The Tarai region is a lowland tropical and subtropical zone with flat alluvial land extending along the Nepal–India border, parallel to the mountains, and there is a fertile, humid, and flat agricultural strip. As shown in Figure 3, forest and cropland are the main land use types in the country. Cropland is mainly distributed in the western and southern regions, and forest land and grassland primarily in the northern and eastern regions.

### 3.2. Agricultural Environment

Two-thirds of Nepal’s population are engaged in agriculture, contributing approximately a quarter of the national GDP. Due to its flat lands, rivers, and fertile soil, most of the agriculture takes place in the Terai area. It can be seen from Figure 4 that since 2000, the cereal production and per land cereal production in Nepal are constantly increasing. Furthermore, the increase in agricultural production technology and conditions of agricultural production have been constantly improving in Nepal. However, with economic development and population growth, cities have expanded unrestrictedly in recent decades. Since 2001, the per capita arable land in Nepal is decreasing. Notably, this puts downward pressure on the future food supply in the country.

Among the vegetal products in Nepal, the main cereal crops are rice, maize, and wheat. From 2000 to 2020, rice production accounted for more than 50% of cereal production, showing a downward trend. The proportion of maize yield was generally stable between 20% and 30%, and the overall growth trend was high. In contrast, wheat production is relatively low and remained below 20% until 2007, but after 2007 the share of wheat production increased. Overall, the structure of cereal crops in Nepal is relatively stable (Figure 5a). Among other vegetal products, the production of sugar, oilseeds, pulses, fruits, and vegetables also showed an increasing trend, with the highest growth recorded in vegetables (1.66 times) and potatoes (1.65 times) (Figure 5b).

Livestock in Nepal are mainly poultry, cattle, sheep, and pigs. Among them, the number of poultry keeps increasing rapidly (3.36 times). The number of cattle and sheep are growing slowly, and the number of pigs is the least. The production of meat, eggs, and milk in Nepal shows an increasing trend. Among them, milk production was the highest and grew the fastest, from 120.30 tons to 245.53 tons. Egg production is the lowest but also continues to increase (Figure 5c).

## 4. Results

### 4.1. Food Consumption and Calorie Supply

#### 4.1.1. Food Consumption and Calorie Supply Based on FAO Data

Figure 6 is a calorie conversion based on FAO food supply data for 2000 and 2020. Over the past 20 years, Nepal’s diet structure has been relatively stable, with grains as the main source of calories, accounting for more than 60%. Food consumption is dominated by plant-based products, with the top five being grains, vegetable oils, roots, beans, and sugars. Eggs accounted for the least consumption in 2020. Grains occupy an absolute position in the dietary consumption as a whole, and the nutritional intake is relatively simple (Figure 6).

#### 4.1.2. Food Consumption and Calorie Supply Based on the Questionnaire

Through interviews and exchanges with the Nepalese residents, we learned about some dietary habits and the main types of food. Nepal has a unique eating habit, usually two meals a day. However, according to the results of the questionnaire, most urban families have adjusted to three meals. Breakfast is relatively simple, mostly milk tea and biscuits. Dinner is usually taken seriously, usually rice with some lentils or curry rice. Rice is one of the main foods in Nepal.

By sorting and summarizing the results of the questionnaire, the research objects are mainly the young and middle-aged population, and the population aged 19–59 makes up about 69.85%. The average daily calorie intake per person in 200 households is 2046 kcal, of which the top three are grains with 1287.49 kcal, meat consumption with 267.56 kcal and beans with 116.46 kcal. Among the main types of food, milk tea and rice are the main foods, accounting for more than 90% (Figure 7). The obtained results after calorie conversion show that the food consumption structure of Nepalese residents is mainly rice, which accounts for 62.91%. This is consistent with the statistical results published by FAO (Figure 6). Meat, vegetables, pulses, and milk are relatively small. The structure of the diet of the Nepalese residents is very simple. Milk tea is the main breakfast, and rice is the main food at noon and in the evening. Grains occupy an absolute position in the dietary consumption as a whole, and the nutritional intake is relatively simple. There is little difference between urban and rural areas.

### 4.2. Spatio-Temporal Characteristics of the Cereal Supply and Demand Balance

According to the Nepal’s cereal consumption in the 2015–2020, the per capita cereal demand in Nepal is 400 kg. This section mainly analyzes the relationship between cereal supply and demand in Nepal from 2000 to 2020 from country to geographical region to county.

(1)Cereal supply and demand balance at the national scale

Figure 8a shows that the carrying capacity of land resources (or LCC) in Nepal maintained a fluctuating increase from 2000 to 2020, but the cereal supply population was consistently lower than the actual population. In 2000, the cereal carrying capacity was 17.79 million people, and the total population of the country was 23.94 million people. The level of cereal supply was far lower than the actual demand of the country. However, with the continuous increase in cereal supply, the carrying population also increases, and the carrying capacity of land resources is significantly increased. By 2020, the gap between the food supply population and the country’s actual population will be significantly narrowed (Figure 8a). Additionally, the average carrying capacity continued to increase (Figure 8b). As a result, the national level of cereal supply has been improving since 2000.

According to the classification of LCCI values (Figure 9), it can also be seen that the national cereal carrying capacity of Nepal is continuously improving. After 2018, it shifted from a deficit to a critical balance. Since 2018, the country has entered a state of tight balance of cereal supply, and the relationship between people and cereal continues to improve. Notably, the level of food supply at the national level has improved significantly.

(2)Cereal supply and demand balance in the geographical regions

Figure 10 shows the supply and demand of cereals in the three geographical regions (i.e., the Mountain belt, the Hill belt, and the Terai belt) in Nepal in 2020. The Mountain region has the lowest carrying capacity with only 1.42 million people, and the Tarai region has the highest carrying capacity of 15.47 million people. The cereal carrying capacity level gradually improved from north to south according to the topographical distribution. Specifically, the Mountain region has harsh climatic conditions, human living environment, and agricultural production conditions, and the entire region has only 1.93 million inhabitants. As a result, the region has the weakest carrying capacity, and food supply and demand are in a state of shortage. Compared to the Mountain region, the Hill region has showed some improvement. The gap between the carrying population and the actual population is the smallest. There are 13.08 million people in the region where the capital is located, accounting for nearly half of Nepal. The current cereal supply cannot meet the population development in the region and is in a state of deficiency. The Tarai region is the main arable land area in Nepal and has excellent conditions for agricultural production, so the food supply is the most abundant, which can meet the needs of about 15.47 million people and is in a state of balance surplus.

(3)Cereal supply and demand balance in the counties

Next, the cereal supply and demand balance in the counties had obvious regional regularities (Figure 11a). The carrying capacity sequentially decreases from the northeast to the southwest. The counties with high carrying capacity are concentrated in the Tarai region, among which Jhapa, Morang, and Kailali have a carrying population of more than 1 million people. The counties with the lowest carrying capacity are concentrated in the Mountains region, such as Manang, Mustang, and Humla with a carrying population of less than 0.03 million people. It can be seen in Figure 11b that the carrying status of Nepalese counties varies greatly. In particular, the carrying index of the capital Kathmandu is 16.92, which is in a state of severe deficiency. The current level of cereal supply can only support 0.15 million people, which is completely unable to meet the local demand of 2.52 million people. In addition, the carrying index of Humla, Bhaktapur, Lalitpur, and other counties is all greater than 2.5, which is in a state of severe deficiency. The gap between the carrying population and the actual population is the smallest in the Hill region, and it is the area with the largest number of counties (14 in total) in the balanced state, especially Jhapa, Bardiya, and Kanchanpur.

### 4.3. Spatio-Temporal Variations of the Calories Supply and Demand Balance

According to the standard of 2400 kcal per capita calorie intake of the Nepalese residents, the change in the calorie supply and demand from country and geographical region to county in the period 2000–2020 was calculated. The number of people who can be satisfied is expressed through the population carrying capacity, and then the relationship of supply and demand and its changes are analyzed in terms of the carrying status.

(1)Calorie supply and demand balance at the national level

The research results in Figure 12a show that, from 2000 to 2020, the carrying capacity of land resources in Nepal has generally maintained a fluctuating increase, and the gap between the carrying population and the actual population is decreasing. Except in 2016, the carrying population exceeded the actual population after 2012. In 2000, the LCC was 22.60 million, slightly lower than the national total population of 23.94 million. By 2020, the LCC will reach 29.78 million people, which is higher than the national 29.13 million. During 2000–2020, the average carrying capacity level also continued to increase (Figure 12b).

Therefore, the level of calorie supply in Nepal is relatively good, ranging from a critical balance to a surplus balance (Figure 13), and has been in a state of balance in the last 20 years. The calorie supply meets the needs of local population development.

(2)Calorie supply and demand balance in the geographical regions

Figure 14 shows the calorie supply and demand in the three geographical regions (i.e., the Mountain belt, the Hill belt, and the Terai belt) in Nepal in 2020, and the results are similar to the cereal supply and demand. The Mountain region has the lowest carrying capacity with only 0.51 million people, while the Tarai region has the highest carrying capacity of 17.40 million people. The calorie carrying capacity gradually improved from north to south in accordance with the topographical distribution. Specifically, the caloric supply is similar to the level of cereal supply, and the Mountain region is still the worst due to the harsh natural conditions and is in a state of deficiency. Compared to the Mountain region, the Hill region has some improvement. The gap between the carrying population and the actual population is the smallest and has increased in relation to the level of cereal supply. The Tarai region is the main arable land area in Nepal, and the conditions of agricultural production are superior, so the level of calories supply is the highest and reaches 17.40 million people, which is in a surplus state overall (Figure 14). Therefore, the supply of calories is much better than that of cereals, but it is still a deficiency in mountainous areas, which may be related to the conditions of local agricultural production and the economic development level.

(3)Calorie supply and demand balance in the counties

It can be seen from Figure 15a that the changes in calorie-based carrying capacity are similar to grains, and it is also similar to the variation in carrying capacity of geographical divisions, with obvious regional laws. The carrying capacity decreases sequentially from the northeast to the southwest. The counties with high carrying capacity are concentrated in the Telai Plain, among which Jhapa, Morang, Kailali, and Bara have a carrying population of more than 1 million people. The counties with the lowest carrying capacity are concentrated in the northern mountainous region. For example, Manang, Mustang, Dolpa, and Mugu have a carrying population of less than 0.02 million people.

Figure 15b shows that the carrying capacity of Nepal’s counties vary widely. The bearing index of the capital Kathmandu is 12.64, which is in a state of severe deficiency. The current level of food supply can only carry 0.20 million people, which is completely unable to meet the local demand of 2.50 million people. In addition, the carrying capacity index of Humla, Bhaktapur, Lalitpur, Kalikot, and other counties are all greater than 2.0, which is in a state of severe deficiency. Combined with the results of cereal supply and demand, the variation law of carrying capacity in Nepal’s counties is relatively consistent, and the spatial distribution is consistent.

## 5. Discussion

### 5.1. Characteristics and Impact of Internal Agricultural Production on Food Supply and Demand

Although the cereal supply and demand balance and the calorie supply and demand balance in Nepal are constantly improving since the 21st century, the level of cereal supply to meet basic human needs is still a long way off. In contrast, a comprehensive supply based on calories can meet the development needs of local population. Therefore, the current food supply can meet the human needs as a whole in Nepal, but there is still a need to continuously improve the level of cereal supply.

Spatially, cultivated land resources are mainly concentrated in the Terai Plain region of Nepal, and the spatial variation in agricultural production capacity is large. By 2020, cultivated land will account for only about 28% of the total land resources in the country, and the cultivated land resources are tight. At the same time, under the influence of population growth and the continuous decline of cultivated land, the cultivated land area per capita is also declining, resulting in an increasingly fragile arable land base for agricultural production, which will have a greater impact on agricultural production, especially food supply [50,51]. In addition, there is a large gap in the level of land resources production in the counties. The basic conditions of agricultural production in the Terai region and the Hill region are good, and the land production capacity is strong, so the pressure on the food supply is relatively low. However, the Mountain region have been affected by natural conditions such as topography and climate, and the food supply is in a state of deficiency.

The growth of food production depends not only on the expansion of planted areas but also on investments in agricultural science and technology, such as good varieties, pesticides, agricultural machinery, and the use of irrigation technology, which require large capital investments [52,53]. At present, the investment of Nepal government in agriculture is insufficient, and the agricultural infrastructure lags behind, which greatly limits the development of agricultural production [54].

### 5.2. Changes in Food Supply and Demand under an Open System

With population growth, economic development, urban expansion, geopolitical cooperation, and integration in Nepal, the supply and demand relations of agricultural products face increasingly complex domestic and foreign situations, and food security must consider domestic supply and international trade. Therefore, the international agricultural market has become an important factor that must be considered to ensure food security objectives. From Nepal’s own supply, its food production cannot meet consumer demand, and by considering food imports and exports combined with the conversion factor, part of the food must be imported. Thus, our study further discusses the supply and demand balance of Nepal’s agricultural products in an open system. Food supply in the open state is expressed by net imports in domestic production. The data originate from the domestic yield, import quantity, and export volume of the main foods in the balance table of FAO.

Figure 16a shows the change in cereal carrying capacity, and we can clearly see that the open system increases Nepal’s cereal carrying capacity before 2010, i.e., from deficiency to severe deficiency. After 2010, the LCCI became small, and it gradually increased the national food carrying capacity. Therefore, moderate food imports are conducive to balanced food, which is conducive to national food security. The supply of own calories can meet the needs of domestic residents, and being under the open system (Figure 16b) is more favorable for the Nepal’s calories supply and demand balance, especially after 2010; i.e., the balanced surplus is turned into a complete surplus. Therefore, international trade is obviously helpful for improving the supply and demand balance of Nepalese agricultural products after 2010. Additionally, moderate import of agricultural products is conducive to national supply and demand balance, and the national development of agricultural products can be appropriately encouraged to ensure domestic food supply and relieve population pressure.

It is worth noting that only the basic foods are considered in this study, but food trade is a complex network that requires further exploration of the impacts of imports and exports on the domestic food supply. For example, individual food types should be analyzed, and in-depth exploration of Nepalese countries in the open system should be performed in order to provide scientific management recommendations for the import and export of domestic food.

## 6. Conclusions

Starting with the premise of single internal food supply, a research framework for regional food supply–demand balance was constructed from the perspectives of land resource utilization, agricultural production characteristics, and the food consumption level and structure. Based on the concept of LCC, this study analyzes and compares the temporal and spatial changes in the food supply and demand balance at the national, geographical, and county scales in Nepal from 2000 to 2020, under the aspects of basic human needs and comprehensive needs. At the same time, changes in food supply and demand in Nepal under the open trade system were discussed. Lastly, this study briefly explored other factors affecting food supply–demand when only considering internal supply and the changes in food supply–demand under the influence of open international trade.

Several key findings emerge from our analysis. First, the production and consumption of agricultural products in Nepal have increased significantly in quantity since 2000. This reflects an increase in agricultural productivity in Nepal and somewhat improved living standards. However, the residents’ nutritional intake is relatively simple, and cereals as the main source of calories account for more than 60%. Second, although the levels of food and calorie supply have fluctuated over the past two decades, the basic level of food supply is far below the actual national demands. Among them, the level of food supply in the plains is the best in the last two decades. In addition, Nepal needs to increase food production to meet the growing needs of its population, and our experiments show that global trade flows of food can greatly improve Nepal’s food supply and demand.

Several important policy implications emerge from our findings. First, it is necessary to ensure the quantity and quality of cultivated land, and improving the quality of cultivated land can directly improve the carrying capacity of land. At the same time, the conclusions of this study can also help the government to understand the development differences between countries and regions in order to increase financial support for areas with weak agricultural development. Second, it is necessary to optimize the food planting structure and improve the nutritional intake status of the residents. We found that Nepal is still on a high grain diet with a serious lack of fat intake. In the future, it is necessary to improve the non-cereal supply capacity and increase the proportion of meat, eggs, and milk supply in the future. In order to achieve the optimization of the food consumption structure, the improvement of the nutrient intake level and the diversification of the nutrient intake structure must be performed. Third, the Nepalese government should focus on improving the level of agricultural cultivation, such as promoting international cooperation in agriculture, increasing funding support for agricultural technology, and improving the management level of crop planting. Of course, appropriate food trade policies can also effectively alleviate the pressure of population development.

This study sheds light on the balance of food supply and demand in Nepal. However, it still has some limitations: First, there are many deficiencies in data, such as the lack of long-term series of sub-county agricultural data. The household consumption questionnaire investigates only some primary foods and cannot well reflect the food consumption of the whole household. Second, research methods can be further improved. In this study, LCC is used to characterize the balance of food supply and demand, while multi-index analysis and footprint method can be used in future studies. Third, our study only considered primary agricultural products, which may have somewhat affected the accuracy of the results. In this regard, the research team will employ other useful methods to collect the necessary data and information to support further research.

## Figures and Tables

**Figure 1 foods-12-01076-f001:**
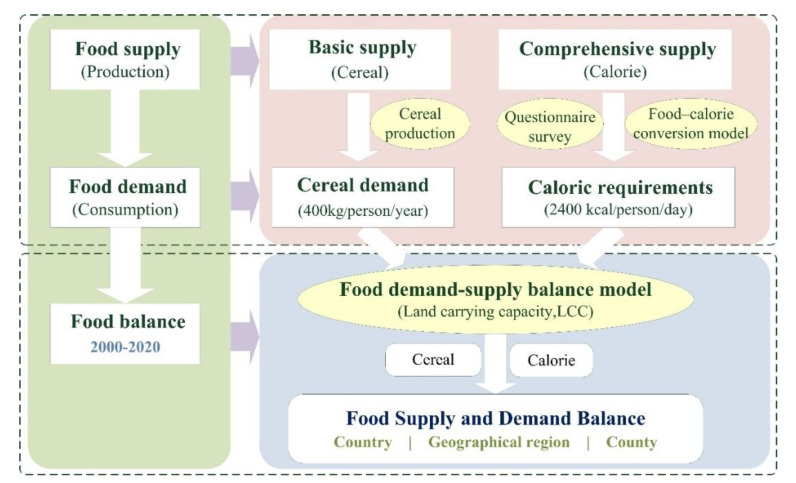
Study framework.

**Figure 2 foods-12-01076-f002:**
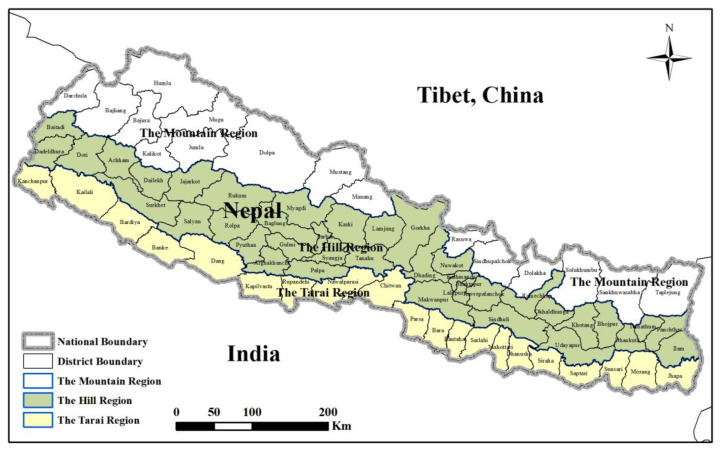
Location and three major ecological divisions of Nepal. Source: *Nepal: A Country Study* [46].

**Figure 3 foods-12-01076-f003:**
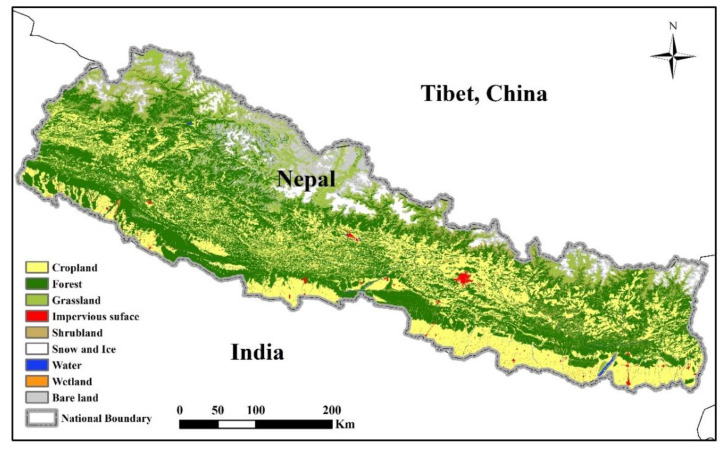
Maps showing land cover types (2020) in Nepal. Source: GlobeLand30 dataset (http://www.globallandcover.com/home_en.html, accessed on 30 September 2022).

**Figure 4 foods-12-01076-f004:**
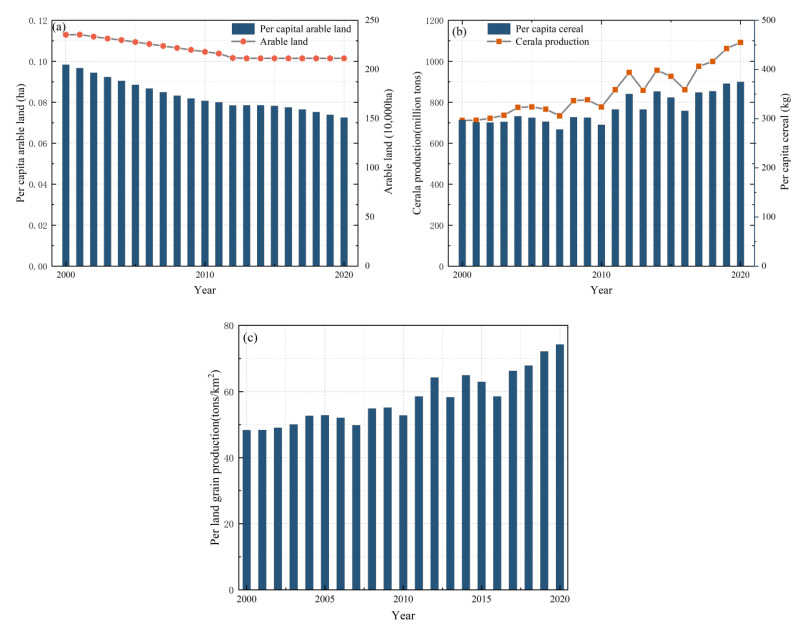
Changes in (**a**) arable land and per capita level, (**b**) cereal production and per capita level, and (**c**) cereal production per land in Nepal from 1990 to 2020.

**Figure 5 foods-12-01076-f005:**
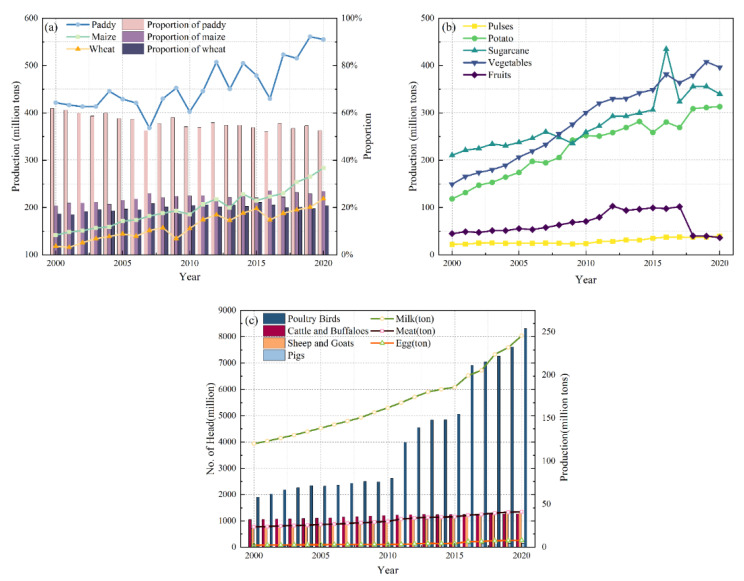
Changes in the production of (**a**) cereal products, (**b**) other vegetal products, and (**c**) animal products in Nepal from 1990 to 2020.

**Figure 6 foods-12-01076-f006:**
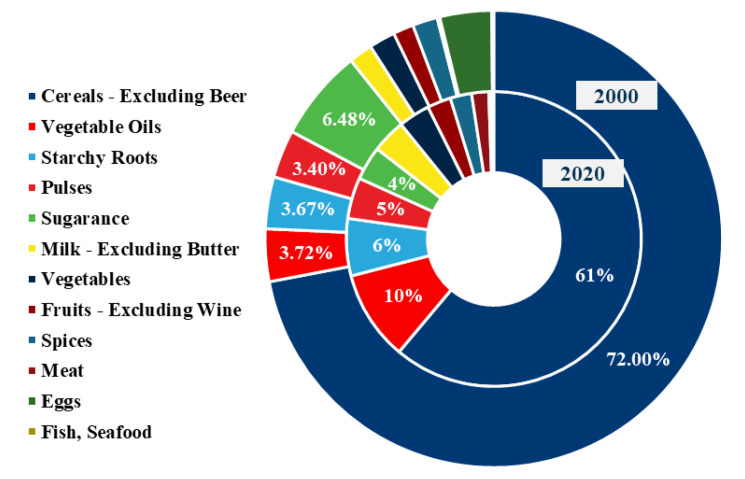
Calorie supply in Nepal in 2000 and 2020 based on FAO data.

**Figure 7 foods-12-01076-f007:**
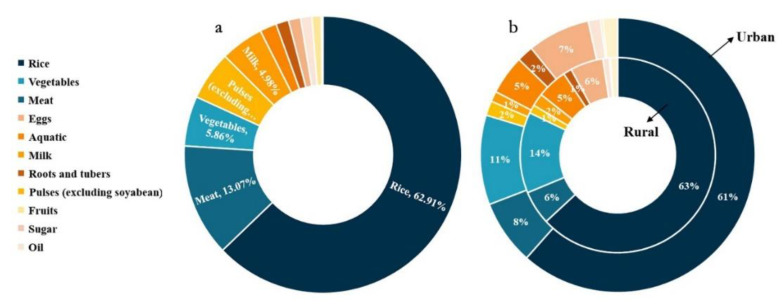
Calorie supply in (**a**) Nepal and (**b**) urban and rural residents in 2020 based on household questionnaire.

**Figure 8 foods-12-01076-f008:**
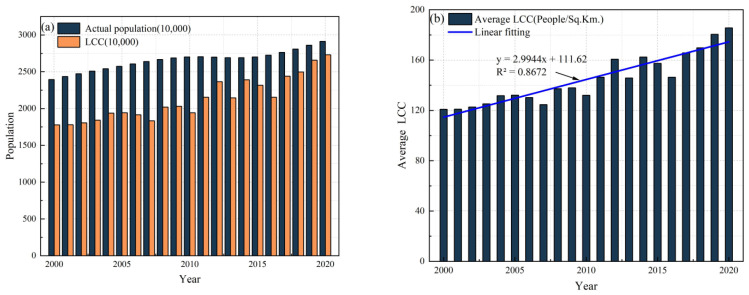
National level of cereal supply in Nepal from 2000 to 2020: (**a**) cereal carrying capacity and real population; and (**b**) average carrying capacity.

**Figure 9 foods-12-01076-f009:**
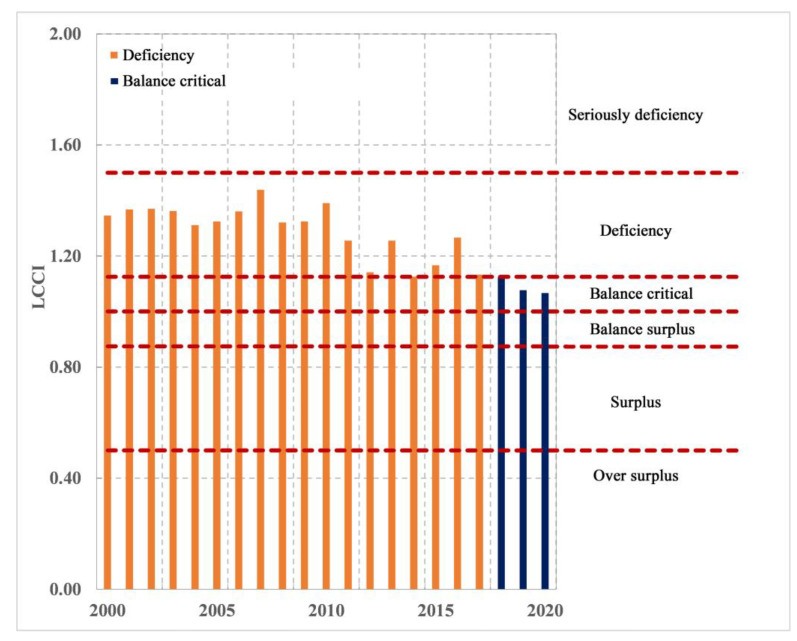
National level of cereal supply and demand from 2000 to 2020.

**Figure 10 foods-12-01076-f010:**
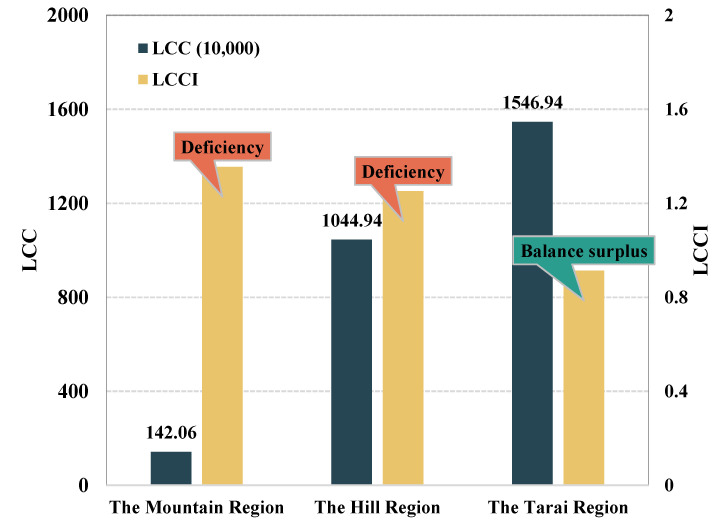
Results of changes in food supply in three geographical regions of Nepal in 2020.

**Figure 11 foods-12-01076-f011:**
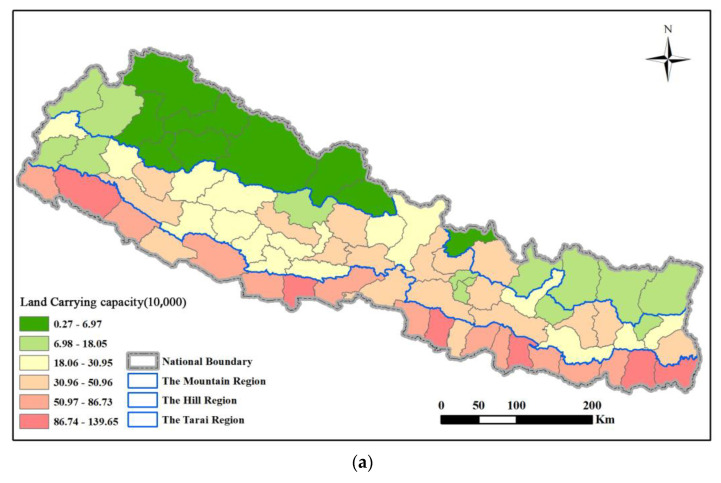
Spatial pattern of (**a**) LCC and (**b**) cereal supply–demand status of counties in Nepal in 2020.

**Figure 12 foods-12-01076-f012:**
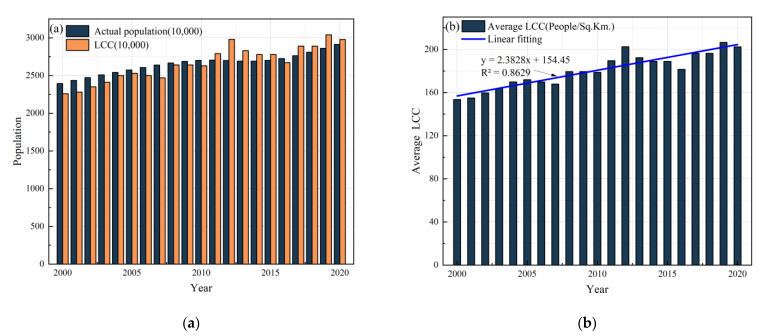
National level of calorie supply in Nepal from 2000 to 2020: (**a**) calories carrying capacity and real population; and (**b**) calorie carrying capacity per land.

**Figure 13 foods-12-01076-f013:**
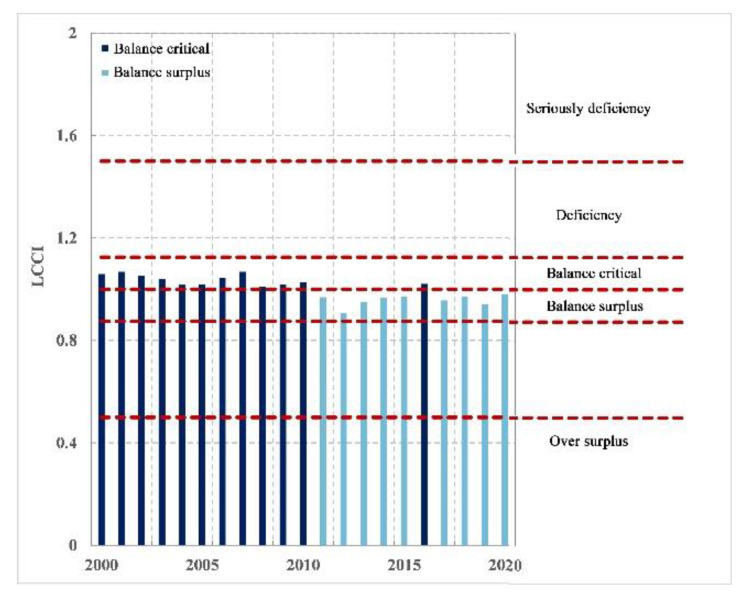
National level of calorie supply and demand from 2000 to 2020.

**Figure 14 foods-12-01076-f014:**
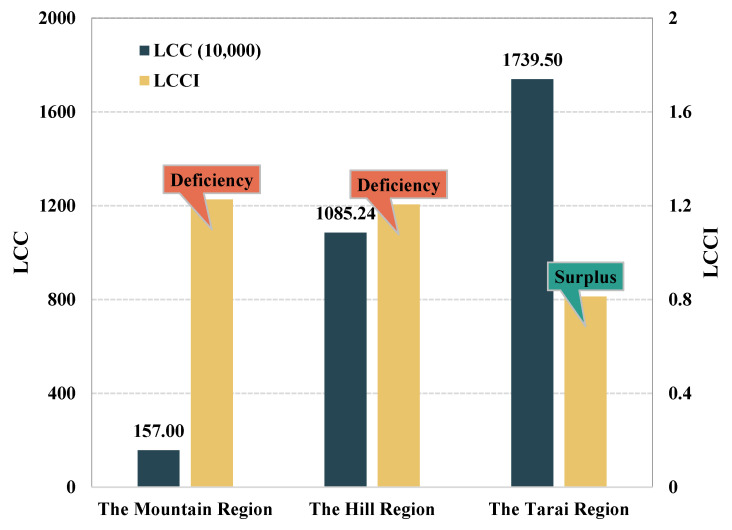
Results of changes in calorie supply in the three geographical regions of Nepal in 2020.

**Figure 15 foods-12-01076-f015:**
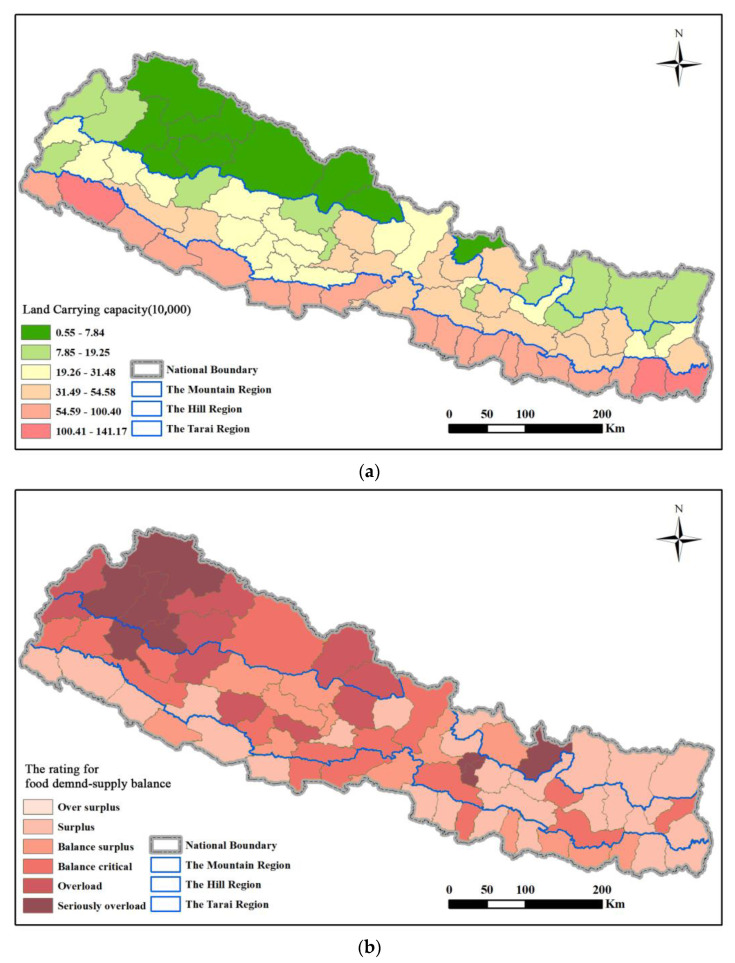
Spatial pattern of (**a**) land carrying capacity and (**b**) calorie supply–demand status of counties in Nepal in 2020.

**Figure 16 foods-12-01076-f016:**
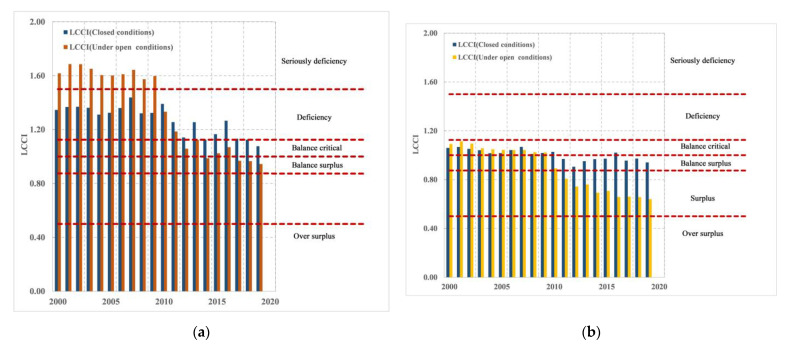
The balance of (**a**) cereal and (**b**) calorie supply and demand in Nepal from 2000 to 2019 according to open and closed system.

**Table 1 foods-12-01076-t001:** Food–calorie conversion parameters for the main food categories.

Food	Calories (kcal/100 g)	Conversion Factor	Intake Part Coefficient	Waste Coefficient
Maize	356	0.78	0.76	0.92
Millet	340	0.78	0.76	0.92
Rice	280	0.9	0.87	0.92
Wheat	334	0.78	0.76	0.92
Potato	67	0.82	0.80	0.76
Sugarcane	30	0.77	0.76	0.88
Pulses	340	0.9	0.89	0.89
Fruits	45	0.77	0.72	0.58
Vegetables	22	0.77	0.72	0.58
Spices	337	0.77	0.72	0.58
Oilseeds	387	0.9	0.89	0.89
Meat	176	1	0.96	0.84
Milk	61	1	0.99	0.87
Egg	139	0.9	0.89	0.87

**Table 2 foods-12-01076-t002:** The evaluation criteria based on LCCI for food demand–supply balance.

Type	Rating	Value Range
Food surplus	Over surplus	<0.5
	Surplus	0.5–0.875
Food balance	Balance surplus	0.875–1.0
	Balance critical	1.0–1.125
Food deficit	Deficiency	1.125–1.5
	Seriously deficiency	>1.5

## Data Availability

The data presented in this study are available request to authors.

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
