# Peer review of "Does Nepal Have the Agriculture to Feed Its Population with a Sustainable Diet? Evidence from the Perspective of Human–Land Relationship"

_foods, 2023, doi:10.3390/foods12051076_

Round 1

Reviewer 1 Report

The manuscript presents interesting original research and I congratulate the authors for this material. However, the manuscript needs to be restructured especially considering the citation sources, as well as the structure of the paper. All these are necessary in order to publish the manuscript. 

Since the citation of the sources, both in the text and the sources of the figures and data in the tables, do not comply with the scientific requirements, but especially with the FOODS journal, I consider it necessary to reformulate, restructure and completely revise the manuscript. In short, I reproduce my point-by-point comments below (these being only a synthesis, because deficiencies appear repeatedly throughout the manuscript), and I also marked them in the PDF format of the manuscript, which I have attached.

1. Section "2. Study area" - Must be presented Materials & Methods section. According to Foods journal "all manuscripts must contain the required sections: Author Information, Abstract, Keywords, Introduction, Materials & Methods, Results, Conclusions, Figures and Tables with Captions, Funding Information, Author Contributions, Conflict of Interest and other Ethics Statements."

2. What are the sources of this information: "As shown in Figure 2, forest and cropland are the mainly land use types in Nepal. 1The cropland is mainly distributed in the western and southern regions, and the forest 136 land and grassland primarily in the northern and eastern regions. "?

3.What is the source of this Figure 1? and the next ones?

4. In Figure 3 (as well as in the text - i.e. row 146) the unit of measurement of the land surface is not expressed. The assumption is that it is about "hectare", but from the formation "Changes in (a) arable land and per capita level, (b) cereal production and per capita level, and (c) per land cereal production in Nepal from 1990 to 2020 " this does not result.

5. It is written that "The agricultural production structure in Nepal is relatively simple, and thus the agricultural products in this study are mainly divided into vegetal and animal products. Vegetal products are divided into eight categories, including grains, roots, sugar crops, oil crops, pulses, spices, vegetables, and fruits. Animal products are divided into three categories, namely meat, eggs, and milk. Agricultural production data are mainly Food 190 and Agriculture Organization of the United Nations (FAO) website (https://www.fao.org/home/en), Statistical Year Book Nepal (2013-2019), Statistical Information on Nepal Agriculture (2019/20). The specific agricultural product data sources are shown in Appendix 1. It should be noted that the same statistical range as FAO was used 194 in this study." The sources must be more specific, more detailed. And must be cited. 

6. How was this questionnaire developed? How was validated? Who reviewed it?

7. "Data on food consumption were obtained from food balance sheet in FAO and house-hold questionnaires. “The sources must be more specific, more detailed. And must be cited.

8. It is write that "Food conversion factor, edible coefficients, and waste coefficients in distribution and 207 consumption are mainly from Global Food Losses and Food Waste[43] " and  this the reference in the References list "43. global food losses and food waste. 2011. " . This is not relevant. 

9. "Land use data in 2020 from GlobeLand30 dataset (http://www.globalland-221 cover.com/home_en.html). The land area data comes from the Statistical Year Book Nepal 222 2019. " -In the text, reference numbers should be placed in square brackets [ ]

10. "The model has parameter matching for each food (Table 1). Table 1 gives the food–calorie conversion parameters for the major categories of foods. For detailed parameters of other foods, please refer to the appendix of the paper. " What is the source of this? Must be cited. 

11. in the Reference list, the calendar dates of accessing the sources must be mentioned (where applicable)

12. References should be described as: 

Author 1, A.B.; Author 2, C.D. Title of the article. Abbreviated Journal Name Year, Volume, page range

Reviewer 2 Report

No specifications are provided about the spatial distribution of the questionnaires and responses. Population conditions of poverty and food availability will, by sure, deeply vary in the different geographic areas.

No considerations are carried about the healthiness of diet and the correct composition; only the calories supplied by cereals and other kind of food is insufficient. Demand should also be studied according to local food culture and the need of higher education about nutrition.

Mountain and hill regions have, of course, a deficit in food production. It is mandatory to evaluate if flatlands can improve their productivity enough to fulfil food requirements of the nation, especially in the consideration of possible future technical improvements and better management of the resources.

Different crop management and cultivated species could help to satisfy the need for quantity, quality and composition of food. The agricultural model to investigate should also consider the environmental impact and the resilience to extreme weather conditions, in order to suggest effective strategies to the government. On this subject very little is said (lines 451-456). Very little is studied about the food trade as well.

Finally, it is mandatory to consider the overall trend of national development: tourism increase, industry, energy, environmental needs. Food and health linkages must be studied in the context of the overall model of Nepalese development path.

Round 2

Reviewer 1 Report

All my observations and recommendations were taken into account in an effective way. The manuscript is much improved, based on the initial original research, and I consider that it can be published as it is now.

Author Response

We are very grateful to the reviewer for their approval and consent for publication.

Reviewer 2 Report

The new release of the study shows very little improvement compared to the former one.

Almost no answer has been given to the questions regarding the sample for the survey and the analyses of the dietary composition in terms of nutritional quality.

No advancement was reported about the topics regarding Nepalese development path and land utilization planning.

In the overall, only English language was significantly improved.
